# Auditory Stimulation Improves Gait and Posture in Cerebral Palsy: A Systematic Review with Between- and Within-Group Meta-Analysis

**DOI:** 10.3390/children9111752

**Published:** 2022-11-15

**Authors:** Shashank Ghai, Ishan Ghai, Susanne Narciss

**Affiliations:** 1Psychology of Learning and Instruction, Department of Psychology, School of Science, Technische Universität Dresden, 01069 Dresden, Germany; 2Centre for Tactile Internet with Human-in-the-Loop (CeTI), Technische Universität Dresden, 01069 Dresden, Germany; 3School of Life Sciences, Jacobs University Bremen, 28759 Bremen, Germany

**Keywords:** paediatric rehabilitation, rhythmic auditory cueing, music-supported therapy, gait, postural stability, sensorimotor learning

## Abstract

The past decade has seen an increased interest in the implementation of auditory stimulation (AStim) for managing gait and postural deficits in people with cerebral palsy. Although existing reviews report beneficial effects of AStim on the spatiotemporal and kinematic parameters of gait, there are still numerous limitations that need to be addressed to correctly interpret these results. For instance, existing reviews have failed to characterize the effects of AStim by conducting separate between and within-group meta-analyses, these reviews have not evaluated the influence of AStim on postural outcomes, and nor have included several high-quality existing trials. In this study, we conducted between- and within-group meta-analyses to establish a state of evidence for the influence of AStim on gait and postural outcomes in people with cerebral palsy. We searched the literature according to PRISMA-P guidelines across 10 databases. Of 1414 records, 14 studies, including a total of 325 people with cerebral palsy, met the inclusion criterion. We report a significant enhancement in gait speed, stride length, cadence, and gross motor function (standing and walking) outcomes with AStim compared to conventional physiotherapy. The findings from this analysis reveal the beneficial influence of AStim on the spatiotemporal and kinematic parameters of gait and postural stability in people with cerebral palsy. Furthermore, we discuss the futurized implementation of smart wearables that can deliver person-centred AStim rehabilitation in people with cerebral palsy.

## 1. Introduction

People with cerebral palsy (CP) exhibit poor spatiotemporal and kinematic control over gait and posture [1,2,3]. This deterioration in gait and postural performance is associated with poorer cognitive [4], social [5], psychological [6], and quality of life outcomes [7,8]. Despite the recent advancements in the field of rehabilitation, poor prognosis is still highly prevalent in people with CP [9,10].

In the past decade, several studies have reported the beneficial effects of auditory stimulation (AStim)-based interventions on gait and postural outcomes in people with CP [11,12,13,14,15,16,17]. These studies suggest that the improvements with AStim are not only limited to the spatiotemporal outcomes of gait (i.e., gait speed, stride length, and cadence) but that they also improve the kinematic outcomes (i.e., 3-dimensional gait analysis of pelvis, hip, knee, and ankle) [13,18,19], which eventually promotes an overall enhancement of gait stability [20]. AStims are defined in the literature as cueing sensorimotor interventions that utilize repetitive isosynchronized beats to elicit motor execution in a synchronized manner with the rhythm [21]. Likewise, other concurrent forms of AStim, such as movement sonification [22,23] and patterned sensory enhancement [24,25], have also been discussed in the existing literature. These types of AStim are mapped on the movement kinematic parameters (e.g., frequency/amplitude mapped with joint angle.) and provide auditory feedback of the executed movement in real time [26,27,28].

Moreover, the published literature have reported that the beneficial impact of AStim is not “condition specific”, as this intervention can facilitate gait and postural performance in other neurologically impaired population groups as well [29,30,31,32,33,34,35]. Studies have mentioned several mechanisms owing to which AStim might instigate improvements in the motor domain. For instance, an increase in the intracortical connectivity between auditory –motor network nodes, especially in the α-, β-, and γ-bands, has been identified as one of the major underlying mechanisms that promote motor recovery [36,37,38,39]. Similarly, reorganization of defunct cortical structures [40], time-locked corticomuscular coherence [41], reduced interhemispheric inhibition [42], and unmasking of existing synapses [43] are other underpinning neurophysiological mechanisms that could also facilitate motor performance in people with CP. In addition, AStim could also aid gait and postural performance in people with CP by simply reducing the extent of cognitive–motor interference [44,45], limiting variability in musculoskeletal co-activations [46], and increasing motivation [13] and arousal [21,47,48] during training.

AStim-based training also possesses the capabilities of facilitating motor performance in a manner that is “patient-oriented” [49,50,51,52]. This approach, in theory, allows the AStim to be mapped on the movement characteristics of a person so that it adjusts according to the preferred pace of the gait or the personalized movement characteristics of the performer [53]. Moreover, this allows the AStim to be delivered according to a person’s choice (i.e., superimposed on the preferred type of music) and, therefore, could allow additional benefits such as active participation and higher motivation from the performer’s perspective [54,55,56]. These properties permit AStim to adhere to the best practice principles of motor rehabilitation, which suggest that interventions should be highly task-specific, intensive, challenging, repetitive, and intriguing to promote recovery [47,57].

To date, two meta-analyses [11,16] have attempted to evaluate the influence of AStim on gait outcomes in people with CP. Within these studies, only one has reported the beneficial effects of AStim on the spatiotemporal parameters of gait from a within-group perspective [16], while the other reported trivial changes in the spatiotemporal parameters of gait with AStim [11]. Likewise, these reviews are limited from the methodological and analytic points of view in several ways. First, these reviews do not account for several existing studies [12,13,14]. This lack of sufficient data could possibly reduce the power of the analyses conducted in the meta-analyses and, in turn, increase the chances that the observed results incurred a type II error [58]. Second, none of the existing reviews evaluated the meta-analysis outcomes from both between- and within-group perspectives. These findings could be important because, while the between-group analyses will explain the differential outcome of AStim as compared to conventional physiotherapy, the within-group analyses would help evaluate the magnitude of change in the gait parameters after the AStim-based intervention, thereby allowing clinicians to deduce appropriate training dosages in their training regimens. Third, none of the included meta-analyses have evaluated the outcomes differentially between randomized controlled trials and controlled clinical trials. Again, this differential analysis would allow for segregation of studies according to the level of bias (i.e., of high and low quality) [59], and, thereafter, their results could be interpreted accordingly. Fourth, no study to date has attempted to evaluate the influence of AStim on the outcome of posture using the gross motor function measure (i.e., subscale D: standing and E: walking), the berg balance scale, and the timed-up-and-go test in people with CP. The evaluation of these outcomes could help in deducing the influence that AStim might have on postural stability outcomes in people with CP. Fifth, no review has differentiated the outcomes of direct AStim (without training) as compared to AStim (with training). The evaluation of this outcome is also important to quantify if training indeed is beneficial for facilitating gait or if only transient AStim could be sufficient in enhancing the gait outcomes in people with CP. Therefore, in this review, we attempt to address these gaps persisting in the current state of the evidence.

In this systematic review and meta-analysis, we conducted a between-group analysis to determine the influence AStim has on spatiotemporal and kinematic parameters of gait and postural stability in people with CP as compared to conventional physiotherapy. Likewise, we also conducted a within-group analysis (i.e., pre vs. post) to evaluate the influence of AStim on spatiotemporal and kinematic parameters of gait and postural stability in people with CP. The rationale for conducting both between- and within-group analyses is largely based on the fact that we want to quantify the magnitude of difference instigated by AStim from two different perspectives. First, from the between-group perspective, we want to demonstrate the differences between the experimental and the control groups. Secondly, we further want to shed light on the magnitude of pre- vs. postchanges induced by AStim on the gait, postural, and kinematic outcomes. We believe that these perspectives can allow clinicians to better understand the overall impact of AStim while simultaneously giving them an opportunity to compare its efficacy with existing rehabilitation interventions. The objectives of this study are as follows:To evaluate the effect of AStim on spatiotemporal gait parameters from both between- and within-group analyses.To evaluate the effect of AStim on postural stability from both between- and within-group analyses.To evaluate the effect of AStim on gait kinematic parameters from both between- and within-group analyses.To perform subgroup analyses between studies according to their training (i.e., no-training vs. training) and randomization (i.e., randomized controlled trials vs. controlled clinical trials) status.

## 2. Materials and Methods

### 2.1. Sources of Data and Search Strategy

PRISMA-P 2020 guidelines were followed to carry out this systematic review and meta-analysis. The checklist has been provided as a Appendix A. This systematic review was preregistered at the Open Science Framework (https://osf.io/xb8zn).

The systematic search of the literature was carried out across 10 databases (Web of Science, PEDro, Pubmed, EBSCO, MEDLINE, Scopus, Cochrane Central Register of Controlled Trials, EMBASE, PROQUEST, and Psychinfo) which were searched from January 1970 to March 2022. The choice of these specific databases was based on the access provided by our institute. The appropriate PICOS search terms were listed in the preregistration protocol. An additional search of the bibliographic section of the eligible studies was also conducted.

The inclusion criteria for the studies were developed according to the SPIDER (sample, phenomenon of interest, design, evaluation, and research type) approach. The inclusion criteria were determined by two researchers (S.G. and I.G.). The studies evaluating the following were included: (1) People with CP; (2) influence of AStim on spatiotemporal (i.e., gait speed, stride length, cadence, stride time, and single/double support duration) parameters of gait in people with CP; (3) studies evaluating the influence of AStim on kinematic parameters of lower limbs during gait (i.e., 3-dimensional analysis of gait deviation index [60]); (4) studies evaluating the influence of AStim on static and dynamic aspects of posture (Berg balance scale [61], timed-up-and-go test [62], gross motor function measure subscale dimension D and E [63]); (5) studies scoring ≥ 4 on the PEDro quality appraisal scale; (6) quantitative studies (except case series, case studies, and review articles); (7) studies published in peer-reviewed academic journals and conference proceedings; (8) studies published in either English, Hindi, German, or French languages. The screening of the abstracts and the full-text articles was independently conducted by two researchers (S.G. and I.G). In the case of discrepancies in the article selection, discussions were held between the two researchers to seek consensus. The following data were extracted from the included articles: authors, country of research, participant information (age, sample size, and gender distribution), gross motor function classification level, evaluated outcomes, training schedule, training groups, auditory cue characteristics, and results. The extracted data have been tabulated in Table 1.

### 2.2. Assessment of the Risk of Bias

The PEDro scale was used to appraise the methodological quality of the included studies [64]. The rating of the PEDro scale (i.e., out of 11) can be interpreted as follows: studies scoring 9–10 were of “excellent quality”, 6–8 were of “good quality”, 4–5 were of “fair quality”, and ≤3 were of “poor quality”. The appraisal of the included studies was carried out by two researchers (S.G. and I.G.) independently.

### 2.3. Data Analysis

In the present review, a between-group (experimental vs. control group) and a within-group (pre- vs. post-AStim), random-effect meta-analysis was conducted with IBM SPSS (V 28.0). The figures for the study were developed using Graphpad Prism software (V 9.3.1). We computed the between-group analysis by using the mean change scores (i.e., pre-/post-performance outcomes) from the respective studies. The data for the meta-analysis were separately distributed and analysed for each spatiotemporal and kinematic outcome of gait (i.e., gait speed, stride length, cadence, and gait deviation index) and each gross motor function measure for standing and walking. The reported outcomes of the meta-analysis included weighted and adjusted effect size (i.e., Hedge’s g), 95% CI, and level of significance. The threshold for the interpretation of effect size is as follows: <0.25 is considered a small effect, ≥0.25 to 0.75 a medium effect, and >0.75 a large effect. Forest plots were generated to illustrate the results. In addition, the heterogeneity in the included studies was quantified by using I^2^ statistics. The threshold for the interpretation of the heterogeneity with I^2^ statistics is as follows: between 0% and 25% is considered as negligible heterogeneity, from 25% to 75% as moderate heterogeneity, and >75% as substantial heterogeneity. In the present study, subgroup analyses were conducted based on study design (i.e., randomized controlled trial and controlled clinical trial) and training status (i.e., AStim-based training and no training). Furthermore, an evaluation of publication bias was conducted according to Duval and Tweedie’s trim-and-fill procedure. The alpha level for the study was set at 5%.

## 3. Results

### 3.1. Characteristics of Included Studies

The initial search across the ten databases yielded a total of 1414 articles, which, after implementing the SPIDER inclusion criteria, were reduced to 14 articles (Figure 1). Thereafter, the qualitative and quantitative data were extracted from the included studies (Table 1). Of the 14 included studies, 6 were randomized controlled trials (RCTs) [14,15,17,18,24,65,66], whereas 8 were controlled clinical trials (CCTs) [12,13,19,67,68,69,70].

**Table 1 children-09-01752-t001:** Details of the included studies.

Authors and Country of Research	Sample Size (N) Gender Distribution (F, M) (Age in Years as Mean ± SD/Range)	Gross Motor Function Classification System Level	Outcomes	Training Schedule	Training Groups with Characteristics	Results
Gerek and Moghadamİ [12] Turkey	N = 12 ?F, ?M (13 ± 3.5)	I	Cadence Gait speed Double support time Single support time Limp index Step width Step length Opposite foot lift	-	AStim at 2/4, 4/4 and 6/8 rhythm	Cadence: Significant ↓ with all AStim (2/4, 4/4, 6/8). Gait speed: Significant ↑ with AStim (6/8) and significant ↓ with AStim (2/4, 4/4). Double support time: Significant ↑ with AStim (4/4, 6/8) and no difference with AStim (2/4). Single support time: Significant ↑ with AStim (2/4, 4/4, 6/8). Limp index: Significant ↑ with all AStim (2/4, 4/4, 6/8). Step width: Significant ↑ with AStim (4/4, 6/8) and no difference with AStim (2/4). Step length: Significant ↑ with AStim (2/4, 4/4) and no difference with AStim (6/8). Opposite foot lift: Significant ↓ with all AStim (2/4, 4/4, 6/8).
Kim, Yoo [13] South Korea	SC: N = 6 4F, 2M (20 ± 2.8) CC: N = 6 5F, 2M (19.5 ± 5.0)	I to II	Cadence Gait speed Stride length Gait kinematics (pelvis, hip, knee) Range of motion (pelvis, hip, knee, ankle)	Session duration: 30 min Times per week: 3 Weeks: 4	SC: Simple chord AStim CC: Complex chord AStim	Cadence: Significant ↑ with AStim (SC, CC). Gait speed: Significant ↑ with AStim (SC, CC). Stride length: Significant ↑ with AStim (SC, CC). Gait kinematics: Significant ↑ in maximal ankle plantar flexion in sagittal plane with AStim (SC, CC). Range of motion: Significant ↑ in ankle ROM with AStim (CC), no difference in ankle range of motion with AStim (SC).
Duymaz [14] Turkey	Exp: N = 60 ?F, ?M (7.42 ± 2.4) Ct: N = 60 ?F, ?M (7.6 ± 2.6)	I to III	Gross motor function measure dimension D and E	Session duration: 45 min Times per week: 3 Weeks: 5	Exp: Classical music AStim with neurodevelopment therapy Ct: Neurodevelopment therapy	Gross motor function measurement D (i.e., standing): Significant ↑ with AStim and neurodevelopment therapy. Gross motor function measurement E (i.e., walking): Significant ↑ with AStim and neurodevelopment therapy.
Ben-Pazi, Aran [15] Israel	Exp: N = 9 1F, 8M (7.7 ± 4.4) Ct: N = 9 4F, 5M (7.1 ± 3.9)	II to V	Gross motor function measure dimension D and E	Session duration: 30 min Times per week: 4 Weeks: 4	Exp: AStim with sound frequencies in gamma range modulated in frequency and/or amplitude embedded in nature sound of preference, children actively listened to tracks and gamma tones faded in the background music/sound over the first two minutes Ct: Nature and music sound according to preference	Gross motor function measurement D (i.e., standing): Significant ↑ with AStim. Gross motor function measurement E (i.e., walking): Significant ↑ with AStim.
Efraimidou, Tsimaras [17] Greece	Exp: N = 5 0F, 5M (35.2 ± 13) Ct: N = 5 0F, 5M (38.8 ± 12.2)	I to II	Timed-up-and-go test 10 m walk test (normal and fast speed) Berg balance scale Centre of pressure sway	Session duration: 50 min Times per week: 2 Weeks: 8	Exp: AStim (70–90 bpm), with 4/4 music meter Ct: Conventional physiotherapy	Timed-up-and-go test: Significant ↑ with AStim. 10 m walk test (normal and fast speed): Significant ↑ with AStim. Berg balance scale: Significant ↑ with AStim. Centre of pressure sway: Significant ↓ with AStim.
Shin, Chong [19] South Korea	N = 7 4F, 3M (30.1 ± 4.1)	-	Cadence Gait speed Stride length Stride time Step time Single support time Double support time Stance/swing phase Gait kinematics (pelvis, hip, knee, ankle, foot) Gait deviation index	Session duration: 30 min Times per week: 3 Weeks: 4	AStim by four-chord progression with metronome beat on keyboard at preferred cadence	Cadence: No difference with AStim. Gait speed: No difference with AStim. Stride length: No difference with AStim. Stride time: No difference with AStim. Step time: No difference with AStim. Single support time: No difference with AStim. Double support time: No difference with AStim. Stance/swing phase: No difference with AStim. Gait kinematics: Significant ↑ in maximal ankle plantar flexion in sagittal plane with AStim. Gait deviation index: No difference with AStim.
Jiang [68] USA	N = 9 5F, 4M (5–12)	I to III	Cadence Gait speed Stride length	Session duration: 30 min Times per week: 1 Weeks: 3	AStim delivered by piano, guitar, bass, percussion, with music in 4/4 beat accentuated by metronome Piano was superimposed on the beat to highlight the rhythm at the participant’s preferred cadence	Cadence: Significant ↑ with AStim. Gait speed: Significant ↑ with AStim. Stride length: No difference with AStim.
Wang, Peng [24] Taiwan	Exp: N =18 6F, 12M (9 ± 1.9) Ct: N =18 3F, 15M (8.9 ± 2.6)	I to III	Gait speed Gross motor function measure dimensions D and E	Exp: Session duration: 25.2 min Number of sessions: 17.8 Ct: Session duration: 26.9 min Number of sessions: 17.7	Exp: AStim (PSE of spatial, temporal and force parameters) Ct: Conventional physiotherapy	Gait speed: No difference with AStim. Gross motor function measurement D (i.e., standing): Significant ↑ with AStim. Gross motor function measurement E (i.e., walking): No difference with AStim.
Varsamis, Staikopoulos [70] Greece	N = 18 7F, 11M (18.2 ± 3.8)	-	Duration for gait performance Number of steps Cadence Number of steps (intra individual standard deviation)	Session duration: 6 min Times per week: 1 Weeks: 1	AStim at preferred cadence	Duration for gait performance: Significant ↑ with AStim. Number of steps: No difference with AStim. Cadence: Significant ↑ with AStim. Number of steps (intra individual standard deviation): Significant ↓ with AStim.
Kim, Kwak [18] South Korea	Exp: N = 15 5F, 10M (27.3 ± 2.4) Ct: N = 15 6F, 7M (27.3 ± 2.5)	-	Cadence Gait speed Stride length Step length Stride time Step time Stance phase Swing phase Gait kinematics (pelvis, hip, knee, ankle, foot)	Session duration: 30 min Times per week: 3 Weeks: 3	Exp: AStim at preferred cadence Ct: Neurodevelopmental/Bobath therapy	Cadence: Significant ↑ with AStim. Gait speed: Significant ↑ with AStim. Stride length: Significant ↑ with AStim. Step length: Significant ↑ with AStim. Stride time: Significant ↓ with AStim. Step time: Significant ↓ with AStim. Swing phase: Significant ↑ with AStim. Gait kinematics: Significant ↑ in pelvic minimal angle of anterior tilt in sagittal plane with AStim. Significant ↓ in pelvic anterior tilt initial contact, max angle of anterior tilt in sagittal plane with AStim. Significant ↓ in hip minimal flexion angle in sagittal plane with AStim. Significant ↓ in hip maximum adduction, abduction angle, and abduction/adduction at initial contact in coronal plane with AStim. Significant ↑ in hip maximum internal rotation in transverse plane with AStim. Significant ↓ in hip maximum external rotation in transverse plane with AStim. Significant ↓ in foot internal/external rotation at initial contact in transverse plane with AStim.
Baram and Lenger [67] Israel	Exp: N =10 6F, 4M (11.1 ± 6.5) Ct: N = 10 7F, 3M (13.3 ± 6.2)	-	Gait speed Stride length	-	Exp: AStim at preferred cadence Ct: Visual cueing	Gait speed: Significant ↑ with AStim. Stride length: Significant ↑ with AStim.
Kim, Kwak [66] South Korea	N = 14 5F, 9M (25.6 ± 7.3)	I to II	Cadence Gait speed Stride length Step length Stride time Step time Stance phase Swing phase Gait kinematics (pelvis, hip, knee, ankle, foot) Gait deviation index	-	AStim at preferred cadence	Cadence: No difference with AStim. Gait speed: No difference with AStim. Stride length: No difference with AStim. Step length: No difference with AStim. Stride time: No difference with AStim. Step time: No difference with AStim. Stance phase: No difference with AStim. Swing phase: No difference with AStim. Gait kinematics: Significant ↓ in pelvic anterior tilt at initial contact in sagittal plane with AStim. Significant ↓ in pelvic maximal, minimal angle of anterior tilt in sagittal plane with AStim. Significant ↓ in hip maximal, minimal flexion angle in sagittal plane with AStim. Significant ↓ in hip internal/external rotation at initial contact in transverse plane with AStim. Gait deviation index: Significant ↑ with AStim.
Hamed and Abd-elwahab [65] Egypt	Exp: N = 15 ?F, ?M (7.03 ± 0.76) Ct: N = 15 ?F, ?M (7.07 ± 0.82)	-	Gait speed Stride length Cadence Gait cycle time	Session duration: 60 min Times per week: 5 Weeks: 12	Exp: Melodious AStim at preferred cadence Ct: Conventional physiotherapy	Gait speed: Significant ↑ with AStim. Stride length: Significant ↑ with AStim. Cadence: Significant ↓ with AStim. Gait cycle time: Significant ↑ with AStim.
Kwak [69] USA	TGT: N = 10 ?F, ?M SGT: N = 10 ?F, ?M Ct: N = 10 ?F, ?M (6–20)	-	Stride length Gait speed Cadence Gait symmetry	Session duration: 30 min Times per week: 5 Weeks: 3	TGT: Therapist guided AStim training SGT: Self-guided AStim training Ct: Conventional physiotherapy	Gait speed: Significant ↑ with AStim. Stride length: Significant ↑ with AStim. Cadence: No difference with AStim. Gait symmetry: Significant ↑ with AStim.

SD: Standard deviation, F: Female, M: Male, SC: Simple chord, CC: Complex chord, Exp: Experimental group, Ct: Control group, PEDI: Paediatric Evaluation of Disability Inventory, ↑: Increase, ↓: Decrease

### 3.2. Risk of Bias

Individual PEDro scoring of each included study has been tabulated in Table 2. The average PEDro quality score of the 14 included studies was 5.9 ± 1.4, suggesting the overall quality of the included studies to be “fair”. Individually, a total of four studies scored 8 on the PEDro score [15,18,24,65], two scored 6 [14,17], seven scored 5 [13,19,66,67,68,69,70], and one scored 4 [12]. The risk of bias scoring across the studies has also been illustrated in Figure 2.

### 3.3. Publication Bias

The incidence of publication bias according to Duval and Tweedie’s trim-and-fill procedure has been demonstrated in Figure 3. The method identified no missing studies on either side of the mean effect. In the analysis, under the random-effect model, the point estimate and the 95% CI for the combined studies were 2.19, −0.562 to 4.95, *p*: 0.11.

### 3.4. Systematic Review Report

#### 3.4.1. Participants

In the included studies, the data from a total of 325 (76F, 97M) people with CP were included. Three of the included studies did not report the gender distribution in their sample [12,14,69]. In addition, two studies reported the age of their sample as a range [68,69]. The average overall age of the included participants was 17.6 ± 10.5 years. In the included studies, data for adults with CP were reported by seven studies (n = 82, age: 25.6 ± 7.1 years). Likewise, eight studies included children with CP (i.e., <18 years, n = 240, age: 9.9 ± 2.7 years).

#### 3.4.2. Gross Motor Function Classification

Eight studies reported the gross motor function classification (GMFC) of their CP cohort. Here, one study included people with GMFC level I [12], three studies included people with a GMFC level between I and II [13,17,66], three studies included people with a GMFC level between I and III [14,24,68], and one study included people with a GMFC level between II and V [15].

#### 3.4.3. Outcome

The qualitative evidence from the present review suggests that AStim is beneficial for enhancing spatiotemporal and kinematic parameters of gait in people with CP. Here, eleven of the included studies reported a significant enhancement (*p* < 0.05) in the spatiotemporal parameters of gait with AStim in people with CP [13,14,15,17,18,65,66,67,68,69,70]. Whereas one study each reported enhancements in the spatiotemporal parameters of gait (*p* > 0.05) without significance [24], no effects of AStim on the outcomes of gait [19], and a negative influence of AStim on the spatiotemporal gait parameters in people with CP [12].

#### 3.4.4. Interventions

The included studies used various types of AStims (see Table 1). Seven of the included studies used AStims that were adjusted according to the preferred cadence of their CP cohort [18,24,65,66,67,68,70]. The rest of the seven studies did not specify if their AStim was adjusted as per the preferred cadence of their participants [12,13,14,15,17,19,69]. In terms of the auditory signal characteristics, one study used AStim at different rhythms (i.e., 2/4, 4/4, 6/8) [12], one study used harmonic modifications of chords (i.e., simple/complex chords) [13], one study merged musical AStim with neurodevelopmental therapy [14], one study used gamma-range frequencies in the music-based AStim [15], one study used AStim at different beats (i.e., 70–90 beats/min at 4/4 music meter) [17], one study used AStims with four chord progression [19], one study used different musical instruments to deliver AStim at a 4/4 beat with a metronome [68], one study used patterned sensory enhancement [24], one study used guided AStim (i.e., self/therapist guided) [69], one study used melodious AStim [65], and four studies did not specify the characteristics of their AStim [18,66,67,70].

### 3.5. Meta-Analysis Report

A detailed meta-analysis report has been provided in Table 3.

**Table 3 children-09-01752-t003:** Meta-analysis outcome.

Outcome Number	Outcome Evaluated	Analysis Type	Number of Studies; (References)	Meta-Analysis Outcome Hedge’s G, 95% Confidence Interval, *p* Value	Heterogeneity Outcome I^2^	Figure
1.	Gait speed	Between-group	N = 5; [17,18,24,65,67]	1.59, 0.71 to 2.47, <0.001	77%	Figure 4A
2.	Gait speed (RCT)	Between-group	N = 4; [17,18,24,65]	1.46, 0.45 to 2.47, 0.004	79%	Appendix A
3.	Gait speed (CCT)	Between-group	N = 1; [67]	-	-	-
4.	Gait speed (with training)	Between-group	Same as outcome number 2			
5.	Gait speed (with training and RCT)	Between-group	Same as outcome number 2			
6.	Gait speed (with training and CCT)	Between-group	None	-	-	-
7.	Gait speed (no training)	Between-group	N = 1; [67]	-	-	-
8.	Gait speed (no training and RCT)	Between-group	None	-	-	-
9.	Gait speed (no training and CCT)	Between-group	N = 1; [67]	-	-	-
10.	Gait speed	Within-group	N = 10; [12,13,17,18,19,24,65,67,68,69]	2.61, −0.72 to 5.96, 0.12	99%	Figure 4B
11.	Gait speed (RCT)	Within-group	N = 4; [17,18,24,65]	1.41, 0.49 to 2.34, 0.003	76%	Appendix A
12.	Gait speed (CCT)	Within-group	N = 6; [12,13,19,67,68,69]	3.64, −2.10 to 9.38, 0.21	100%	Appendix A
13.	Gait speed (with training)	Within-group	N= 8; [13,17,18,19,24,65,68,69]	4.06, −0.44 to 8.57, 0.07	99%	Appendix A
14.	Gait speed (With training and RCT)	Within-group	Same as outcome number 11			
15.	Gait speed (With training and CCT)	Within-group	N = 4; [13,19,68,69]	7.61, −2.58 to 17.81, 0.14	100%	Appendix A
16.	Gait speed (no training)	Within-group	N = 2; [12,67]	−1.01, −2.68 to 0.64, 0.23	92%	Appendix A
17.	Gait speed (no training and RCT)	Within-group	None	-	-	-
18.	Stride length	Between-group	N = 3; [18,65,67]	2.27, 1.68 to 2.86, <0.001	0%	Figure 5A
19.	Stride length (RCT)	Between-group	N = 2; [18,65]	2.07, 1.43 to 2.72, <0.001	0%	Appendix A
20.	Stride length (CCT)	Between-group	N = 1; [67]	-	-	-
21.	Stride length (with training)	Between-group	Same as outcome number 19			
22.	Stride length (with training and RCT)	Between-group	Same as outcome number 19			
23.	Stride length (with training and CCT)	Between-group	N = 1; [67]	-	-	-
24.	Stride length	Within-group	N = 7; [13,18,19,65,67,68,69]	3.51, −0.79 to 7.82, 0.11	99%	Figure 5B
25.	Stride length (RCT)	Within-group	N = 2; [18,65]	1.97, −0.06 to 4.01, 0.05	89%	Appendix A
26.	Stride length (CCT)	Within-group	N = 5; [13,19,67,68,69]	4.39, −1.98 to 10.77, 0.17	99%	Appendix A
27.	Stride length (with training)	Within-group	N = 6; [13,18,19,65,68,69]	4.08, −1.09 to 9.26, 0.12	99%	Appendix A
28.	Stride length (with training and RCT)	Within-group	Same as outcome number 25			
29.	Stride length (with training and CCT)	Within-group	N = 4; [13,19,68,69]	5.23, −2.54 to 13.0, 0.18	100%	Appendix A
30.	Stride length (with no training)	Within-group	N = 1; [67]	-	-	-
31.	Stride length (with no training and RCT)	Within-group	None	-	-	-
32.	Stride length (with no training and CCT)	Within-group	N = 1; [67]	-	-	-
33.	Cadence	Between-group	N = 2; [18,65]	0.51, −2.77 to 3.80, 0.76	97%	Figure 6A
34.	Cadence (RCT)	Between-group	Same as outcome number 33			
35.	Cadence (CCT)	Between-group	None	-	-	-
36.	Cadence (with training)	Between-group	Same as outcome number 33			
37.	Cadence (with training and RCT)	Between-group	Same as outcome number 33			
38.	Cadence (with training and CCT)	Between-group	None	-	-	-
39.	Cadence (with no training)	Between-group	None	-	-	-
40.	Cadence	Within-group	N = 8; [12,13,18,19,65,68,69,70]	−0.70, −1.72 to 0.30, 0.17	93%	Figure 6B
41.	Cadence (RCT)	Within-group	N = 2; [18,65]	−0.14, −2.38 to 2.09, 0.90	94%	Appendix A
42.	Cadence (CCT)	Within-group	N = 6; [12,13,19,68,69,70]	−0.83, −2.01 to 0.35, 0.16	93%	Appendix A
43.	Cadence (with training)	Within-group	N = 7; [13,18,19,65,68,69,70]	0.10, -0.46 to 0.67, 0.71	72%	Appendix A
44.	Cadence (with training and RCT)	Within-group	Same as outcome number 41			
45.	Cadence (with training and CCT)	Within-group	N = 5; [13,19,68,69,70]	0.15, −0.37 to 0.69, 0.56	55%	Appendix A
46.	Cadence (with no training)	Within-group	N = 1; [12]	-	-	-
47.	Cadence (with no training and RCT)	Within-group	None	-	-	-
48.	Cadence (with no training and CCT)	Within-group	N = 1; [12]	-	-	-
49.	Gait deviation index	Between-group	None	-	-	-
50.	Gait deviation index	Within-group	N = 4; [13,18,19,66]	0.61, −0.15 to 1.39, 0.11	69%	Appendix A
51.	Gait deviation index (RCT)	Within-group	N =1; [18]	-	-	-
52.	Gait deviation index (CCT)	Within-group	N = 3; [13,18,19]	0.26, −0.22 to 0.74, 0.28	0%	Appendix A
53.	Gait deviation index (with training)	Within-group	Same as outcome number 52			
54.	Gait deviation index (with training and RCT)	Within-group	N = 1; [18]	-	-	-
55.	Gait deviation index (with training and CCT)	Within-group	Same as outcome number 52			
56.	Gait deviation index (with no training)	Within-group	None	-	-	-
57.	GMFM-D	Between-group	N = 3; [14,15,24]	0.50, 0.20 to 0.81, <0.001	0%	Appendix A
58.	GMFM-D (RCT)	Between-group	Same as outcome number 57			
59.	GMFM-D (CCT)	Between-group	None	-	-	-
60.	GMFM-D (with training)	Between-group	Same as outcome number 57			
61.	GMFM-D (with training and RCT)	Between-group	Same as outcome number 57			
62.	GMFM-D (with training and CCT)	Between-group	None	-	-	-
63.	GMFM-D (with no training)	Between-group	None	-	-	-
64.	GMFM-D	Within-group	N = 3; [14,15,24]	0.44, 0.14 to 0.74, 0.004	0%	Appendix A
65.	GMFM-D (RCT)	Within-group	Same as outcome number 64			
66.	GMFM-D (CCT)	Within-group	None	-	-	-
67.	GMFM-D (with training)	Within-group	Same as outcome number 64			
68.	GMFM-D (with training and RCT)	Within-group	Same as outcome number 64			
69.	GMFM-D (with training and CCT)	Within-group	None			
70.	GMFM-D (with no training)	Within-group	None	-	-	-
71.	GMFM-E	Between-group	N = 3; [14,15,24]	0.24, −0.05 to 0.53, 0.11	0%	Appendix A
72.	GMFM-E (RCT)	Between-group	Same as outcome number 71			
73.	GMFM-E (CCT)	Between-group	None	-	-	-
74.	GMFM-E (with training)	Between-group	Same as outcome number 71			
75.	GMFM-E (with training and RCT)	Between-group	Same as outcome number 71			
76.	GMFM-E (with training and CCT)	Between-group	None	-	-	-
77.	GMFM-E (with no training)	Between-group	None	-	-	-
78.	GMFM-E	Within-group	N = 3; [14,15,24]	0.78, −0.33 to 1.90, 0.17	88%	Appendix A
79.	GMFM-E (RCT)	Within-group	Same as outcome number 78			
80.	GMFM-E (CCT)	Within-group	None	-	-	-
81.	GMFM-E (with training)	Within-group	Same as outcome number 78			
82.	GMFM-E (with training and RCT)	Within-group	Same as outcome number 78			
83.	GMFM-E (with training and CCT)	Within-group	None	-	-	-
84.	GMFM-E (with no training)	Within-group	None	-	-	-

RCT: Randomized controlled trial, CCT: Controlled clinical trial, GMFM: Gross motor function measure, GMFM-D: Subgroup score for evaluating standing, GMFM-E: Subgroup score for evaluating walking.

## 4. Discussion

The aim of this systematic review and meta-analysis was to synthesize the current state of knowledge regarding the influence of AStim on spatiotemporal and kinematic parameters of gait and postural stability in people with CP. The findings from the between-group meta-analysis suggest a positive influence of AStim on all the outcomes of gait and posture. In terms of the within-group analysis, a large effect enhancement in the spatiotemporal gait parameters of gait speed and stride length, but not cadence, was observed. Likewise, positive within-group changes were also reported for lower-limb kinematics (i.e., gait deviation index) and postural stability (i.e., gross motor function measure subscales D and E) with AStim.

To date, only two previously conducted meta-analyses have quantified the influence of AStim on spatiotemporal parameters of gait [11,16]. Here, the first review, among eight studies, reported trivial enhancements with AStim only for the outcome of gait speed (i.e., mean difference: 0.03) in people with CP [11]. In contrast, the second review, among nine studies, reported within-group enhancements in spatiotemporal outcomes of gait speed (g: 1.1), stride length (g: 0.5), and cadence (g: 0.3) [16]. In the present study, we, extend the findings of these previous reviews and report enhancement in gait and postural outcomes with AStim in 14 studies. This is in line with the existing literature, wherein the evaluation of spatiotemporal parameters of gait has been emphasised to justify both transient- and training-related changes in gait speed [71,72]. We report significant increments for the outcomes of stride length and cadence, which in turn translated to enhancements in gait speed. However, in the between-group analysis of the spatiotemporal parameters, we observed differences in the overall magnitude of change in the variables (i.e., gait speed: 1.59, stride length: 2.27, cadence: 0.51). Similarly, from the within-group perspective, these changes were again discrepant (i.e., gait speed: 2.61, stride length: 3.51, cadence: −0.70). Three major reasons might explain these differences. Firstly, a huge difference persisted in the number of studies that included between- (i.e., gait speed: five, stride length: three, cadence: two) and within-group (i.e., gait speed: 10, stride length: seven, cadence: eight) analyses. Here, the reduction in the number of studies could have perhaps reduced the power of the analyses, hence increasing the possibility of bias [73]. Secondly, we observed a strong discrepancy in terms of the designs of studies included in our analysis. For instance, in the between-group analysis of gait speed, only five studies (i.e., one CCT and four RCTs) reported the outcome between control and experimental groups, whereas 10 studies (i.e., six CCTs and four RCTs) reported the outcome of gait speed from a within-group perspective. This discrepancy in the number of studies could be an additional reason why the gait outcomes were different in the within- and between-group analyses. Thirdly, as the step length and cadence account for the modulation in gait speed, it might be possible that medium effect increments in the cadence and large effect increments in stride length for the between-group analysis resulted in larger magnitude increments in gait speed. This might also be the case for the within-group analyses, wherein a reduction in cadence and increased stride length were reported alongside an increase in the gait speed outcome.

The existing literature proposes several mechanisms that could explain the increments in the gait and postural performance of people with CP. For instance, several studies have reported that AStim can facilitate motor outcomes primarily because of its capability to instigate learning in a manner that is task-specific, challenging, motivational, immersive, and multisensory [21,37,47,57]. Under such circumstances, people with CP who are often restricted in their capabilities to learn and perform motor tasks due to shortfalls in their sensory domains (i.e., audition and proprioception) might benefit [74,75]. Hamed and Abd-elwahab [65] affirmed the increase in spatiotemporal gait parameters with AStim to the repetitive and goal-directed nature of training AStim provoked. The authors mentioned that AStim-based training might have allowed their CP cohort to train in an organized, meaningful, and concentrated manner because of the functional reorganization or the neural plasticity it promoted [76]. Although not evaluated by the studies included in our review, neurophysiological studies by laboratories across the world have shown that AStim can augment the deficit perceptual mechanisms commonly observed in people with CP, thereby aiding the performer to execute and learn a task easily [38,39,53,77]. Furthermore, the enrichment of the defunct sensory pathways with training could then allow people with CP to forge robust internal models (i.e., feedforward and feedback loop) needed to facilitate long-term potentiation. A neuroimaging study by Schmitz et al. [53] suggested that training with AStim can enhance the training-related benefits by possibly amplifying the activity in the action observation systems and the parts of the motor loop. The authors reported pronounced activity in the superior temporal sulcus, bilateral insula, precentral gyri, and parts of the temporal regions (superior, medial, and posterior). Likewise, neuroimaging studies exclusively evaluating AStim have reported augmented activation in the deficit cortical and subcortical structures [78,79]. A recent case series highlighted that, in addition to amplified neuroimaging activity, training with an auditory–motor task can aid in establishing robust cortico–cortical connections between the auditory–motor network nodes (See Figure 7 [36]. The authors reported strong coherence, especially in the α- and β-band of the human brain rhythms. In the present case, we hypothesize that perhaps AStim augmented the activation of deficit neural structures and improved intracortical connectivity, which eventually facilitated gait performance in people with CP (i.e., unmasking the pre-existing inactive motor representations).

In addition to the spatiotemporal parameters, an augmentation in gait speed translates to an improvement in gait kinematic patterns. In a study by Lamontagne and Fung [80], the authors reported that an augmentation in gait speed could result in improvements in muscle activation patterns, which could improve lower-limb kinematics. Oudenhoven et al. [81] reported that, in children with CP, faster gait speeds resulted in an improvement by approximately 5° in the hip and knee extension during the stance phase. In this study, the findings are confirmed as we report an increase in the gait deviation index outcome (g: 0.61) together with an increase in gait speed. Another study by Kim et al. [13], in this review, stated an increase in the gait speed together with an increase in the range of motion at the ankle joint and the angle of plantarflexion during the preswing phase. The authors also mentioned that, regardless of the type of AStim (i.e., simple or complex chord), they observed an increase in hip extension during the terminal stance. The authors suggested that this increase in extension could be associated with the forward motion (i.e., push-off) during the initial swing phase of the gait, which is negatively affected in people with CP [82]. Moreover, another study in our review reported an increase in pelvic anterior tilt and hip flexion during the gait cycle together with an increase in gait speed [18]. Furthermore, a few of the included studies in this review also reported enhancements in postural stability with AStim [14,15,17,24]. Efraimidou et al. [17], for instance, reported an improvement in the Berg balance score (Exp: 49.8 ± 5.6 vs. Ct: 42 ± 4.3) and a reduction in the timed-up-and-go test (Exp: 7.5 ± 1.4 vs. Ct: 9.8 ± 1.4 s) in the group training with AStim as compared to conventional therapy. Another study by Ben-Pazi et al. [15] reported beneficial effects of AStim on the gross motor function measure during standing (mean difference: Exp: 5.1 vs. Ct: −0.6). The authors suggested that perhaps the use of auditory stimulation in the γ-frequency could have stimulated cortical activity, synchronized oscillatory neural activity, and/or amplified GABAergic inhibitory neurotransmission in their cohort training with AStim. In this study, the analysis confirms the beneficial influence of AStim on the beneficial outcome of postural stability, as both between- (g: 0.50) and within-group (g: 0.44) increases in the gross motor function measure outcomes for standing in people with CP were observed.

Additionally, we made an attempt to compare the influence between studies evaluating the influence of AStim-based training and simple AStim without training. Previously published reviews had not evaluated this outcome and yet it is important to quantify if training is indeed necessary to avail enhancements in motor outcomes or whether transient AStim could be sufficient as well. The present analysis reports that AStim-based training was far superior in terms of enhancing gait speed (i.e., train: 1.75 vs. no-train: −1.02). These findings are consistent with the existing evidence suggesting that a sufficient amount of auditory–motor training is necessary to establish auditory–motor coupling. For instance, Bangert and Altenmüller [83], in a longitudinal direct-voltage electroencephalography evaluation emphasized that intensive auditory–motor training is essential in eliciting intermodal auditory–sensorimotor co-activation. The authors mentioned that although these co-activations emerge just minutes (i.e., 20 min) after the training, a prolonged training (i.e., 5 weeks) is nonetheless necessary to consolidate the co-activations and for establishing a robust audio–motor-interfaced map. Collectively, the findings from this meta-analysis update and extend the state of knowledge provided by the existing reviews, with the inclusion of 14 studies (n = 325). The present analysis provides further novel evidence regarding the beneficial influence of AStim on gross motor function outcomes, as well as reports the separate influence based on the design (i.e., RCTs and CCTs) and training status (i.e., AStim with and without training) of the studies.

### 4.1. Limitations

Few limitations were present in this study. The main objective of this study was to evaluate the influence of AStim on spatiotemporal and kinematic parameters of gait and postural stability in people with CP. However, upon deeper inspection of the included studies, we observed that while some studies had evaluated the direct influence of AStim, others had evaluated the influence of AStim-based training on the outcomes of gait and posture. Although we carried out separate subgroup analyses to characterize the differences in the influence of AStim-based training and simple AStim, there were still discrepancies in the studies included in these analyses. For instance, studies evaluating the influence of AStim-based training differed largely in terms of the dosage of training duration (for details see Table 1). In this study, AStim-based training interventions were more beneficial as compared to simple AStim, but which specific training dosage was the most superior was not evident. Moreover, the studies included in this review also differed in terms of the implementation of AStim. Here, while some studies had provided Astim at the preferred cadence of the participant, others had not. Similarly, heterogeneity also existed in terms of the characteristics of the auditory signals, i.e., some studies had embedded the AStim on the music [15,17], some studies had used a simple metronome [12,67,70], and some mapped the cue on the spatiotemporal parameters of participant’s gait [24,65]. In our personal opinion, we believe it is time that future studies aptly standardize the AStim-based interventions regarding the “auditory signal characteristics”, the relevant training dosages, and how they try to implement the intervention in the context of rehabilitation. Another major limitation of this study was that fewer studies were included in the within-group analyses of cadence (i.e., two studies), stride length (i.e., three studies), and gross motor function measures (i.e., three studies for both standing and walking subsets). Likewise, some of the included studies in the analysis had a very small sample size. Although we had computed the weight of the effect size according to the sample size of each study (i.e., smaller sample was allocated lesser weight and vice versa) [84], we cannot rule out the possibility of incurring a type II error. The reader is therefore requested to infer these results with caution.

### 4.2. Future Directions

Although the number of studies incorporating AStim for gait rehabilitation in people with CP has increased in the past decade, there are still a few aspects that warrant research. For instance, limited research has evaluated the influence of real-time AStim on gait and postural outcomes in people with cerebral palsy [22,24]. One such intervention is movement sonification. Sonification typically allows the transposition of kinematic parameters of movement with auditory signal characteristics in real time. The feedback stimuli then produced, in principle, could allow the facilitation of motor perception and performance by specifically targeting the neural networks associated with biological motion perception [53]. Previously conducted neuroimagery studies have conclusively demonstrated that passively listening to congruent sonified human actions can improve the timing of movement and also facilitate the auditory–motor entrainment effect [53]. This could be mainly due to the close proximity of the stimuli to the biological motion, which in turn activates the human action observation network. In addition, behavioural studies have also demonstrated that sonification can facilitate proprioceptive accuracy [85,86], and help in synchronizing cyclic movement patterns in a manner that discrete AStims (such as rhythmic auditory cueing) are unable to do [50,87]. Perhaps, training with movement sonification can allow people with CP to better perceive their own movement patterns and determine appropriate movement amplitudes in order to perform their gait effectively.

Furthermore, during the literature search, we observed that the implementation of smart wearable equipment in the rehabilitation of CP is discussed rarely. We believe that the avenue of smart wearables has a huge potential for the motor rehabilitation of people with CP. Smart wearables, in principle, can allow for tracking the activity of a person while simultaneously delivering treatment in a tailored manner. Recently conducted interdisciplinary studies have conceived the notion of using a smart kinesio tape [88,89]. These smart kinesio tapes have inbuilt sensors and actuators and have been reported to actively measure the spine kinematics of a performer and transform it instantaneously in movement sonification. This type of multidisciplinary collaborative approach can allow for actively combining the benefits of two different approaches [90]. For instance, while sonification could facilitate the perception of the activity being conducted through the auditory domain, the kinesio tape could facilitate the tactile mechanoreceptors and augment the perception of the executed movement from an additional afferent stream [91]. Several studies have already demonstrated the beneficial effects of kinesio tapes on gait and postural outcomes in people with cerebral palsy [92,93]. The application of such smart wearables could also allow for the delivery of rehabilitation that is tailored to a person’s needs and thereby could be an important step in the direction of facilitating motor rehabilitation in people with CP.

## 5. Conclusions

The present meta-analysis reports the positive influence of AStim on kinematic, spatiotemporal parameters of gait and postural stability in people with CP. The increments for gait speed and stride length were confirmed to be higher in the meta-analysis when compared to conventional gait rehabilitation by physiotherapy. In addition, subgroup analyses revealed that AStim-based training resulted in a higher increment in gait speed as compared to AStim without training. The present findings should be inferred with caution as the included studies were of “fair” methodological quality and high heterogeneity was observed in the principal meta-analyses. Overall, the present study recommends the incorporation of AStim-based training in gait rehabilitation protocols for people with CP.

## Figures and Tables

**Figure 1 children-09-01752-f001:**
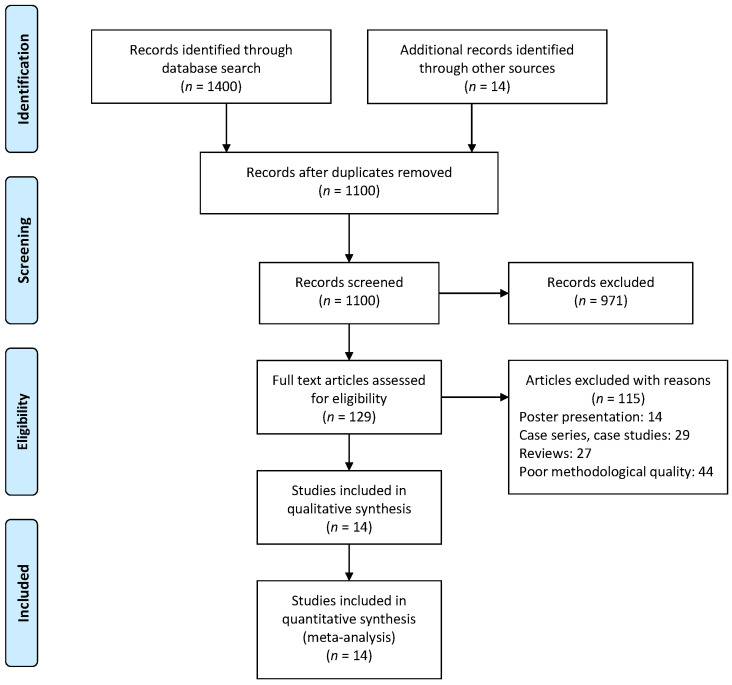
PRISMA flowchart.

**Figure 2 children-09-01752-f002:**
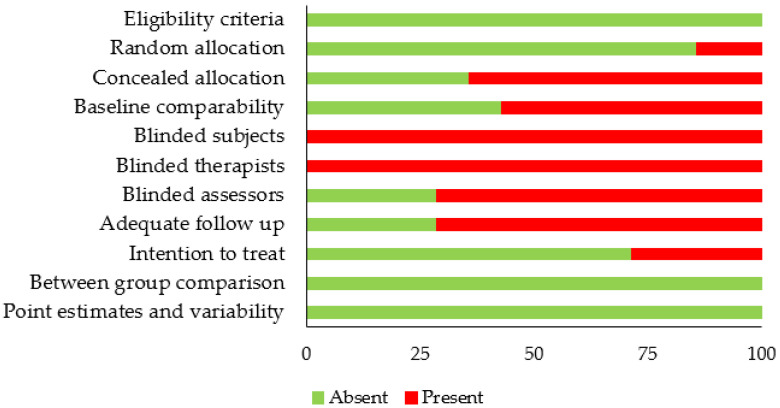
Risk of bias according to the PEDro scale (Absent: Risk of bias absent, Present: Risk of bias present).

**Figure 3 children-09-01752-f003:**
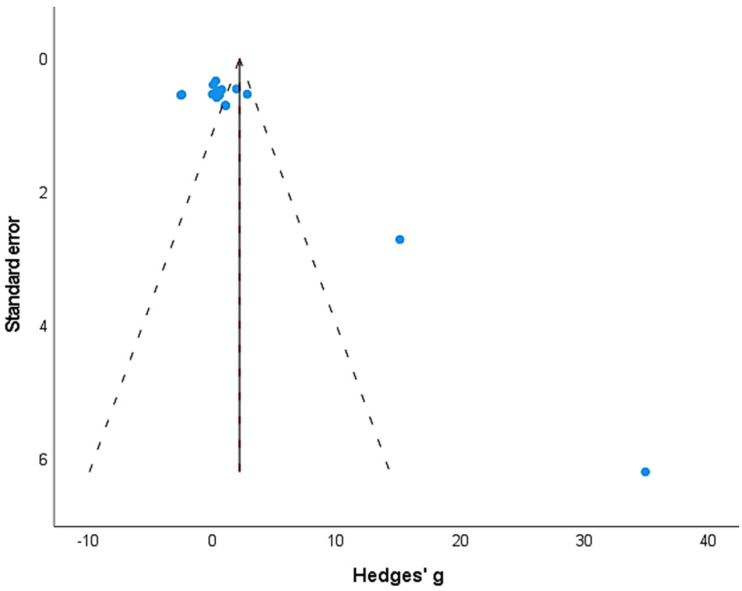
A trim-and-fill funnel plot illustrating the publication bias. Each study is represented by an individual blue circle. The funnel plot area covers 95% of the pseudoconfidence intervals. The vertical midline represents the estimated overall effect size (i.e., observed + imputed studies).

**Figure 4 children-09-01752-f004:**
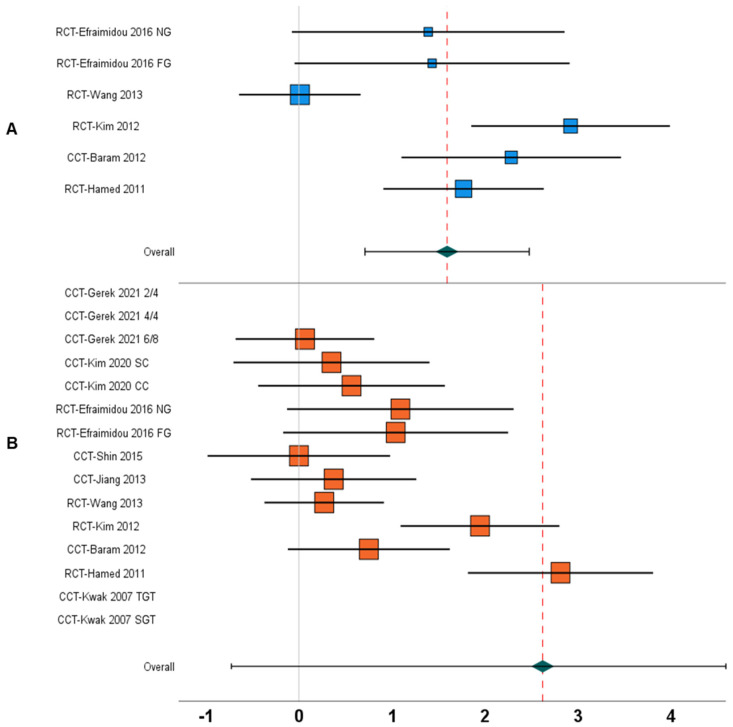
Forest plot illustrating the effect of AStim on gait speed in people with cerebral palsy. The forest plots incorporate individual weighted effect size (Hedge’s g), which is represented as boxes (blue: between-group, orange: within-group), whereas the 95% confidence intervals are represented with whiskers. At the bottom, the pooled weighted effect size and 95% CI are represented with a green diamond. (**A**) In this analysis, a positive overall effect size means an enhancement in gait speed for the AStim group, whereas a negative overall effect size indicates enhancements in gait speed for the control group. (**B**) A positive overall effect size means an enhancement in gait speed with AStim, whereas a negative overall effect size indicates reduction in gait speed with AStim (RCT: Randomized controlled trial, CCT: Controlled clinical trial, NG: Normal gait, FG: Fast gait, TGT: Therapist-guided training, SGT: Self-guided training).

**Figure 5 children-09-01752-f005:**
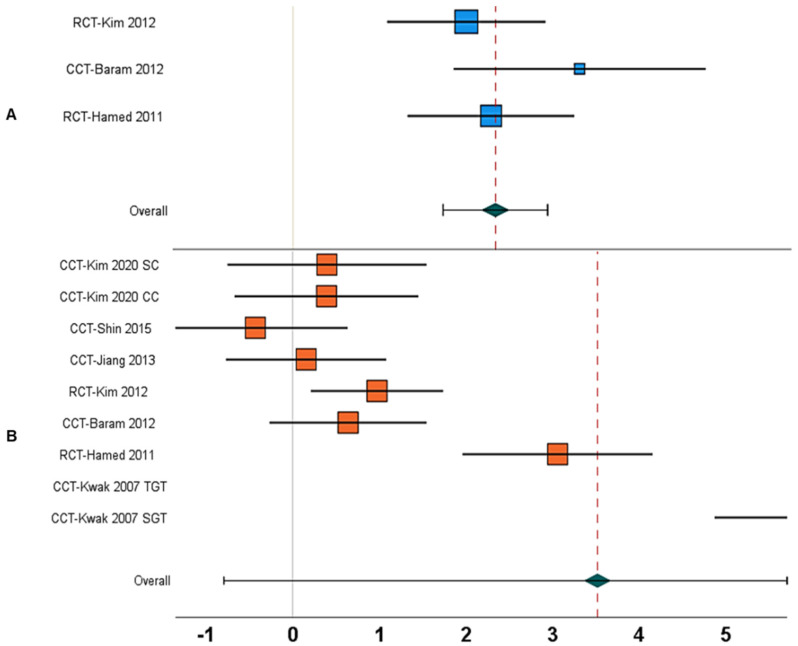
Forest plot illustrating the effect of AStim on stride length in people with cerebral palsy. The forest plots incorporate individual weighted effect size (Hedge’s g), which is represented as boxes (blue: between-group, orange: within-group), whereas the 95% confidence intervals are represented with whiskers. At the bottom, the pooled weighted effect size and 95% CI are represented with a green diamond. (**A**) In this analysis, a positive overall effect size means an enhancement in stride length for the AStim group, whereas a negative overall effect size indicates enhancements in stride length for the control group. (**B**) A positive overall effect size means an enhancement in stride length with AStim, whereas a negative overall effect size indicates reduction in stride length with AStim (RCT: Randomized controlled trial, CCT: Controlled clinical trial, NG: Normal gait, FG: Fast gait, TGT: Therapist-guided training, SGT: Self-guided training).

**Figure 6 children-09-01752-f006:**
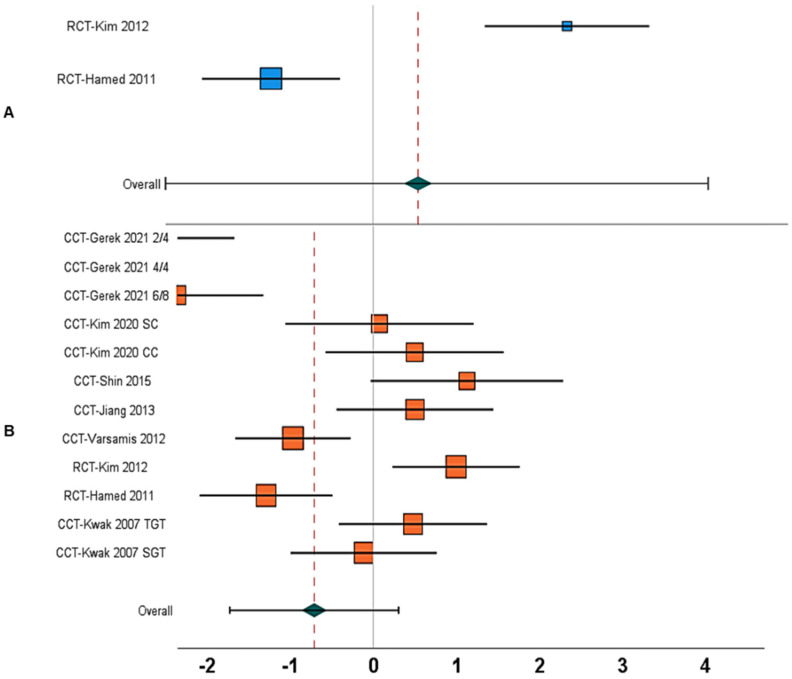
Forest plot illustrating the effect of AStim on cadence in people with cerebral palsy. The forest plots incorporate individual weighted effect size (Hedge’s g), which is represented as boxes (blue: between-group, orange: within-group), whereas the 95% confidence intervals are represented with whiskers. At the bottom, the pooled weighted effect size and 95% CI are represented with a green diamond. (**A**) In this analysis, a positive overall effect size means an enhancement in cadence for the AStim group, whereas a negative overall effect size indicates enhancements in cadence for the control group. (**B**) A positive overall effect size means an enhancement in cadence with AStim, whereas a negative overall effect size indicates reduction in cadence with AStim (RCT: Randomized controlled trial, CCT: Controlled clinical trial, NG: Normal gait, FG: Fast gait, TGT: Therapist-guided training, SGT: Self-guided training).

**Figure 7 children-09-01752-f007:**
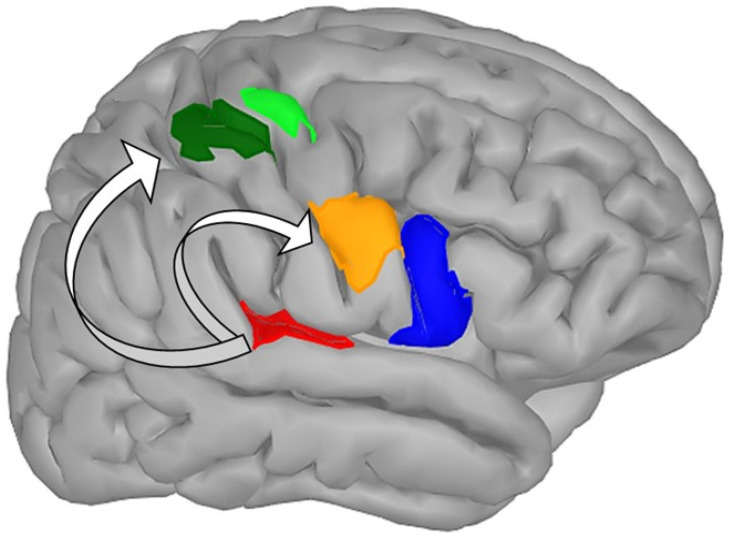
Auditory–motor training amplifies intracortical connectivity between auditory–motor network nodes (Red: auditory cortex, orange: premotor cortex dorsal, blue: inferior frontal gyrus, green: motor cortex, light green: premotor cortex ventral).

**Table 2 children-09-01752-t002:** Detailed PEDro scoring of the included studies (1: bias absent, 0: bias present).

	PEDro score	Point estimates and variability	Between-group comparison	Intention to treat	Adequate follow up	Blinded assessors	Blinded therapists	Blinded subjects	Baseline comparability	Concealed allocation	Random allocation	Eligibility criteria
Gerek and Moghadamİ [12]	4	1	1	1	0	0	0	0	0	0	0	1
Kim, Yoo [13]	5	1	1	1	0	0	0	0	1	0	0	1
Duymaz [14]	6	1	1	1	0	0	0	0	1	0	1	1
Ben-Pazi, Aran [15]	8	1	1	0	1	1	0	0	1	1	1	1
Efraimidou, Tsimaras [17]	5	1	1	1	0	0	0	0	0	0	1	1
Shin, Chong [19]	5	1	1	1	0	0	0	0	0	0	1	1
Jiang [68]	5	1	1	1	0	0	0	0	0	0	1	1
Wang, Peng [24]	8	1	1	0	1	1	0	0	1	1	1	1
Varsamis, Staikopoulos [70]	5	1	1	1	0	0	0	0	0	0	1	1
Kim, Kwak [18]	8	1	1	0	1	1	0	0	1	1	1	1
Baram and Lenger [67]	5	1	1	1	0	0	0	0	0	0	1	1
Kim, Kwak [66]	5	1	1	1	0	0	0	0	0	0	1	1
Hamed and Abd-elwahab [65]	8	1	1	0	1	1	0	0	1	1	1	1
Kwak [69]	5	1	1	1	0	0	0	0	0	0	1	1

## Data Availability

The data can be provided upon reasonable request.

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
