# Peer review of "Auditory Stimulation Improves Gait and Posture in Cerebral Palsy: A Systematic Review with Between- and Within-Group Meta-Analysis"

_children, 2022, doi:10.3390/children9111752_

Round 1
Reviewer 1 Report
This study conducted meta-analysis on the papers investigating the effect of the Astim. This paper is well written. Please address the following two concerns.
In some studies included in this analysis, sample size was very small. Are those appropriate to include for conducting meta-analysis? Please state the rationale of this issue.
The authors conducted meta-analysis even on the studies with within-group data analysis. As far as I know, meta-analysis is usually conducted on the studies using RCT. Please state the rationale of this issue.
Author Response
Comment 1: In some studies included in this analysis, sample size was very small. Are those appropriate to include for conducting meta-analysis? Please state the rationale of this issue.
Response: We thank the reviewer for this critical comment. Indeed, some studies included in this review had a small sample size, but since our goal was to include as many studies as possible we did not exclude these studies. Moreover, regarding the appropriateness of inclusion in meta-analysis, we would like to mention that the weight of the effect for each studies included in the analysis was dependent upon the sample size (1). Therefore, any reduction in sample size meant that lesser weight was allocated to that study and vice versa for the studies with larger sample sizes. Despite these explanations, we do agree with the concern of the reviewer, that the very small sample size in some of the included studies might raise concern regarding the reliability and validity of our results. Therefore, now we have discussed this in the limitations section of the manuscript.
Page 28-29, Line 450-458: Another main limitation of this study was that fewer studies were included in with-in-group analyses of cadence (i.e., two studies), stride length (i.e., three studies), and within-group analysis of gross motor function measures (i.e., three studies for both standing and walking subsets). Likewise, some of the included studies in the analysis had a very small sample size. Although we had computed the weight of the effect size according to the sample size of each study (i.e., smaller sample was allocated lesser weight and vice versa), we cannot rule out the possibility of incurring a type II error. The reader is therefore requested to infer these results with caution.
Comment 2: The authors conducted meta-analysis even on the studies with within-group data analysis. As far as I know, meta-analysis is usually conducted on the studies using RCT. Please state the rationale of this issue.
Response: We thank the reviewer for this important comment. Indeed, meta-analyses are usually conducted on RCTs and in this present study we did follow this procedure. Initially we had conducted joint analyses with different study designs and then later in the sub-group analyses, we do report outcomes separately for RCTs and CCTs. The rationale for conducting both between and within-group analyses was largely based on the fact that we wanted to quantify the magnitude of difference instigated by AStim from two different perspectives. First, in the between-group perspective we demonstrated the differences between the experimental and the control group. Secondly, we tried to then shed further light on the magnitude of pre vs. post changes induced by AStim on the gait and postural outcomes. We believe that these perspective can allow clinicians to better understand the overall impact of AStim while simultaneously giving them an opportunity to compare its efficacy with existing rehabilitation interventions.
Page 3, Line 108-116: The rationale for conducting both between and within-group analyses is largely based on the fact that we want to quantify the magnitude of difference instigated by AStim from two different perspectives. First, from the between-group perspective, we want to demonstrate the differences between the experimental and the control group. Secondly, we further want to shed light on the magnitude of pre vs. post-changes induced by AStim on the gait, postural, and kinematic outcomes. We believe that these perspectives can allow clinicians to better understand the overall impact of AStim while simultaneously giving them an opportunity to compare its efficacy with existing rehabilitation interventions.
Reviewer 2 Report
The present study aimed to systematically review the influence of AStim on gait and postural stability in CP by conducting a within- and between meta-analysis. The authors found a positive influence of AStim on gait and postral stabilility in CP.
In general the aim and scope of the meta-analysis is interesting, I do have some concerns.
One of the extra added value of the meta-analysis compared to previous reviews is the aim to investigate the inlfuence of AStim on postural stability. However, only 3 papers discuss postural stability outcomes. Therefore results of might be written with more caution.
Lastly, I was wondering how AStim can be implemented in CP with GMFCS level V and how these results might give information about improvements in gait parameters (study ref 15, outcome GMFM E) given that participants with GMFCS level V are transported in a wheelchair in all settings. Can extra information about the therapy content be added in table 1?
Author Response
Comment 1: One of the extra added value of the meta-analysis compared to previous reviews is the aim to investigate the inlfuence of AStim on postural stability. However, only 3 papers discuss postural stability outcomes. Therefore results of might be written with more caution.
Response: We thank the reviewer for this important comment. We agree with the thought of the reviewer that only three articles discussing the postural stability outcomes might not be sufficient in explaining the overall effect of AStim on postural stability and that the reader must interpret the results with caution. We have now mentioned this in the limitation section of our manuscript.
Page 28-29, Line 450-458: Another major limitation of this study was that fewer studies were included in with-in-group analyses of cadence (i.e., two studies), stride length (i.e., three studies), and within-group analysis of gross motor function measures (i.e., three studies for both standing and walking subsets). Likewise, some of the included studies in the analysis had a very small sample size. Although we had computed the weight of the effect size according to the sample size of each study (i.e., smaller sample was allocated lesser weight and vice versa), we cannot rule out the possibility of incurring a type II error. The reader is therefore requested to infer these results with caution.
Comment 2: Lastly, I was wondering how AStim can be implemented in CP with GMFCS level V and how these results might give information about improvements in gait parameters (study ref 15, outcome GMFM E) given that participants with GMFCS level V are transported in a wheelchair in all settings. Can extra information about the therapy content be added in table 1?
Response: We thank the reviewer for pointing this out this critical aspect. Unfortunately, the study by Ben-Pazi et al. did not mention how improvements in gait parameters were quantified in CP cases with GMFCS level V. Nonetheless, we have included all the additional information about the therapy content for both AStim group and control therapy in Table 1.
The changes are mentioned on Page 8.
Reviewer 3 Report
This review on an important and different subject is valuable even if the number of publications on the subject is small. Written language and methodology are suitable.
Author Response
Comment: This review on an important and different subject is valuable even if the number of publications on the subject is small. Written language and methodology are suitable.
Response: We thank the reviewer for this constructive comment. We understand the reviewer’s concern that the number of publications included in this review are less. We have now included this in the limitation section of our manuscript.
Page 28-29, Line 449-458: Another main limitation of this study was that fewer studies were included in with-in-group analyses of cadence (i.e., two studies), stride length (i.e., three studies), and within-group analysis of gross motor function measures (i.e., three studies for both standing and walking subsets). Likewise, some of the included studies in the analysis had a very small sample size. Although we had computed the weight of the effect size according to the sample size of each study (i.e., smaller sample was allocated lesser weight and vice versa), we cannot rule out the possibility of incurring a type II error. The reader is therefore requested to infer these results with caution.